# Extraction of Bioactive Compounds from the Fruits of Jambolan (*Syzygium cumini* (L.)) Using Alternative Solvents

**DOI:** 10.3390/plants13152065

**Published:** 2024-07-26

**Authors:** Carla Adriana Ferrari Artilha-Mesquita, Ana Paula Stafussa, Patrícia Daniele Silva dos Santos, Oscar de Oliveira Santos, Silvio Claudio da Costa, Grasiele Scaramal Madrona

**Affiliations:** 1Postgraduate Program in Food Science, Universidade Estadual de Maringá, Avenida Colombo 5790—Zona 7, Maringá 87020-900, PR, Brazil; c.artilha@yahoo.com.br (C.A.F.A.-M.); anastafussa@gmail.com (A.P.S.); 2Chemistry Department, State Universidade Estadual de Maringá, Avenida Colombo 5790—Zona 7, Maringá 87020-900, PR, Brazil; patriciadanieless@hotmail.com (P.D.S.d.S.); oosjunior@uem.br (O.d.O.S.); 3Biochemistry Department, State University of Maringá (UEM), Avenida Colombo 5790—Zona 7, Maringá 87020-900, PR, Brazil; sccosta@uem.br; 4Food Engineering Department, State Universidade Estadual de Maringá, Avenida Colombo 5790—Zona 7, Maringá 87020-900, PR, Brazil

**Keywords:** anthocyanins, ecofriendly, jambolan, food analysis, new technologies

## Abstract

This work demonstrates the effectiveness of using alternative solvents to obtain jambolan extracts with a high content of bioactive compounds compared to conventional organic solvents, being the first study to evaluate the best ecological solvent alternative for *Syzygium cumini* (L.) Skeels. Five alternative solvents were used for extraction: water at 25 °C (W25), water at 50 °C (W50), water at 75 °C (W75), water with citric acid at 2.4% (CA2), and water with citric acid at 9.6% (CA9) in comparison with three conventional solvents: ethanol (EtOH), water with ethanol at 50% (WE), and water with methanol at 50% (WM). A protocol was then established for the extraction and concentration of samples obtained with these solvents. The highest content of total phenolic compounds (TPCs) in the extracts was obtained with the solvent W75 (1347.27 mg GAE/100 g), while in the concentrates it was the solvents EtOH (3823.03 mg GAE/100 g) and WM (4019.39 mg GAE/100 g). Total monomeric anthocyanins (TMAs) increased by 209.31% and 179.95% in extractions with CA2 and CA9, respectively, compared to pulp (35.57 mg eq c-3-g/100 g), demonstrating that they are the most efficient alternative solvents in this extraction. The levels of bioactive compounds and antioxidant activity varied according to the solvents used. Delphinidin 3,5-diglucoside, cyanidin 3,5-diglucoside, delphinidin 3-glucoside, petunidin 3,5-diglucoside, cyanidin 3-glucoside, peonidin 3,5-diglucoside, malvidin 3,5-diglucoside, petunidin 3-glucoside, and malvidin 3-glucoside were identified in most of the samples by UPLC-MS/MS. This study suggests that a simple procedure using alternative solvents can be used as an environmentally friendly strategy to achieve efficient extraction of bioactive compounds in jambolan.

## 1. Introduction

The fruits of the genus *Syzygium* are mentioned in some studies due to the quality of their phytochemical composition and because they have numerous beneficial health properties, which are related to antibacterial, antiprotozoal, antiviral, antifungal, antidiabetic, anti-inflammatory, anticancer, and antiseptic activities [1].

In this genus, jambolan (*Syzygium cumini* (L.) Skeels) appears as an important source of antioxidants, as it has high levels of anthocyanins and high amounts of total phenolic compounds, making it a functional food [2,3,4] and health promoter. Its sensory characteristics are striking and cause a great visual impact because it has an exotic flavor due to the mixture of acidity, astringency, and sweetness, in addition to having a deep purple pulp due to the presence of anthocyanins, which are hydrophilic pigments with antioxidant and anticancer bioactivity, and which have the advantage of high solubility in aqueous mixtures [3,5].

According to the authors Kowalska et al. [6] and López et al. [7], these natural pigments are highly unstable in the presence of light, high temperatures, and pH, and their natural extracts are susceptible to these degradations. Because of this, a cautious analysis must be considered when evaluating the extraction of biologically active substances from this group, since their solubility and stability will depend on the extraction techniques, the conditions performed, and the solvents used.

Most methods of extracting bioactive compounds use traditional techniques using organic solvents, such as acetone, methanol, ethanol, and propanol or mixtures thereof, since these solvents demonstrate excellent dissolution and extraction capacity; however, most of these solvents have intrinsic drawbacks, such as high toxicity, non-biodegradability, cost, and accumulation in the atmosphere due to low boiling points [8]. In addition, the solubility of bioactive compounds in a specific solvent is a peculiar characteristic of the phytochemicals present in the evaluated matrix, which justifies the lack of a universal procedure for their determination [9].

In this sense, given the polar nature of the substances present in jambolan, the use of alternative solvents, such as pure water and acidified water, as sustainable techniques for the extraction of biologically active substances from the fruit, and that propose a reduction in the amount of waste generated by the end of reactions, substitution, reduction, or elimination of toxic reagents and/or solvents, or that generate toxic products and by-products and that guarantee the obtaining of products of adequate quality and safety [7,10], is a proposal for effective, alternative, and ecologically friendly chemical processes.

In view of the above, the use of alternative solvents as sustainable techniques for the extraction of bioactive compounds present in jambolan creates options for obtaining extracts rich in bioactive compounds that can be used in food matrices.

In addition to the fact that very few studies provide information on the definition of the best extractor solvent of jambolan bioactive compounds, this is the first study with the aim of evaluating the best alternative ecological solvent. Thus, the objective of this work is to evaluate the extraction and concentration of the extracts rich in bioactive compounds from jambolan, using alternative solvents in comparison with conventional solvents.

## 2. Results and Discussion

### 2.1. Characterization of Jambolan Pulp

Table 1 describes the centesimal composition and physical–chemical parameters of the jambolan pulp; it is noted that the values corroborate data from the literature. The jambolan fruit is described as a fruit with a remarkable flavor: sweet, acidic, and astringent, with pH ranging from 3.22 to 4.15, acidity at 0.65 to 0.89 g citric acid/100 g, and solids content between 13° to 19° Brix [4,11,12].

Pereira et al. [13] and Vital et al. [14] described jambolan pulp, respectively, with a content of 79.50 g/100 g and 83.96–85.96 g/100 g of moisture, 0.41 g/100 g and 0.30–0.39 g/100 g of ashes, 0.97 g/100 g and 0.69–0.81 g/100 g of proteins, 0.35 g/100 g and 0.23–0.30 g/100 g of lipids, and 17.96 g/100 g and 12.57–13.94 g/100 g of total carbohydrates. The content of nutritional compounds, phytochemicals, and the physical–chemical composition present in the fruit pulp must be close to the fruit in natura and are important to guarantee and maintain a quality standard of an established formulation during its processing, avoiding unnecessary adjustments, and, in general, the results obtained in the centesimal and physical–chemical composition were similar to the literature (mentioned above), indicating that jambolan has potential for industrial processing.

Regarding the bioactive compounds, it was observed that there is a heterogeneity in the values; this difference can be attributed to the varied methodologies or even the variability of the fruits. In this study, the content of total phenolic compounds (TPCs) in the edible part of the fruit was higher in relation to the results reported by Branco et al. [15], Coelho et al. [16], and Vital et al. [14], who found, respectively, 206.95 mg/100 g, 321.90 mg/100 g, and 416.82–573.89 mg/100 g in fresh weight (f.w.) and similar to the result of Madani et al. [3], with 786 mg/100 g (f.w.).

Similarly, the content of total flavonoids (TFs) was higher when compared to the results reported by Coelho et al. [16], with 15.83 mg/100 g in a sample of frozen pulp jambolana, and Branco et al. [15], with 25.29 and 29.45 mg/100 g in a sample of jambolan pulp with and without pasteurization in fresh weight (f.w.). According to Rufino et al. [17], fruits are classified into three categories regarding the content of total phenolic compounds: low (<100 mg GAE/100 g), medium (100–500 mg GAE/100 g), and high (>500 mg GAE/100 g) for samples based on fresh matter. Thus, we can classify jambolan as a fruit with high polyphenolic content.

In our experiments, the content of total monomeric anthocyanins (TMAs) was about six times lower than the values reported by Nascimento-Silva et al. [4], with 210 mg c-3-g/100 g, and also Branco et al. [15], with 213 mg c-3-g/100 g. When comparing the anthocyanins content found in this study with other fruits that also stand out for containing this compound, such as açaí (22.8 mg/100 g), strawberry (23.7 mg/100 g) [18], and camu-camu (10.23 mg/100 g), the latter belonging to the same family as jambolan [19], it appears that the anthocyanin content obtained in this work without the interference of solvents and extraction techniques was higher than these fruits and is within the range of values pointed out by Lestario et al. [20], which ranged from 28.5 to 1318.4 mg/100 g (f.w.). It should be noted that the variability between the results may be due to differences between cultivars, growing conditions, differences in ripening stages, extraction methods, and analyses [3,20].

Regarding the antioxidant capacity of jambolan, the fruit demonstrated excellent antioxidant potential and was superior to the results reported by Rufino et al. [17], who found 173 μM eq Fe_2_SO_4_/g (FRAP method) and 10.49 μM TE/g (ABTS method) for jambolan juice, and Coelho et al. [16], who reported 21.67 μM TE/g in frozen jambolan pulp (ABTS method). The antioxidant capacity by the DPPH method obtained an inhibition percentage of 47.54% (158.69 μM TE/g) higher than the value indicated by Ghosh et al. [21] of 31.29%, and about 12 times higher for frozen jambolan pulp (12.29 μM TE/g) and 18 times higher for jambolan juice (8.48 μM TE/g), respectively [16]. In this study, jambolan pulp proved to be a fruit rich in bioactive nutrients and with high antioxidant capacity, which can be consumed fresh or used in processing for various formulations. In addition, it has been recognized as a nutraceutical fruit due to the presence of powerful antioxidant compounds such as ascorbic acid, anthocyanins, and total phenols [22].

### 2.2. Extraction of Bioactive Compounds with Different Types of Alternative Solvents

The results showed that the contents of TPCs, TFs, and TMAs varied significantly (*p* < 0.05) among the samples analyzed and according to the type of solvent used (Table 2). Extraction of phenolic compounds with water proved to be an efficient alternative solvent when compared to conventional solvents (EtOH, WE, and WM). With the heating of the water in the extractions, the W50 and W75 samples increased the polyphenol content, favoring their extraction by 53.73% and 64.06%, respectively, compared to the sample W25 extract and increased by up to 89.35% in relation to the TPCs content of the jambolan pulp. This occurs because the increase in temperature can help the extraction of phytochemical compounds and also in their transfer from the peel to the pulp. As the temperature rises, the enzymes that degrade the phenolic compounds are inactivated, preserving these compounds in the matrix [23]. The same behavior was also observed by Andrade et al. [24] with the extraction of phenolic compounds in guava pulp (100 °C, 35 min) and Machado et al. [23] in fruits and vegetables (nectarine, black plum, strawberry, red cabbage, eggplant, mango, kiwi, blackberry, and red lettuce) subjected to different temperatures (50 °C, 75 °C, and 100 °C; 1, 2, 5, 10, and 15 min), which showed an increase in the content of phenolic compounds. On the other hand, samples of aqueous extracts when heated decreased by up to 95.83% (W50) and 93.36% (W75) in relation to the TFs content of jambolan pulp and 22.09% (W50) and 31.62% (W75) in relation to the TMAs content of jambolan pulp, demonstrating that under these conditions, these alternative solvents are less efficient for extracting these compounds. According to Andrade et al. [24], increasing temperature can cause logarithmic destruction of anthocyanins; however, this will depend on the structure of anthocyanins in the face of thermal degradation, as is the case of acylated (stable) anthocyanins and non-acylated (unstable) anthocyanins.

Anthocyanins are mainly located in the peel of jambolan fruits, and the W25 extract obtained a higher concentration (about 2.3 times) when compared to jambolan pulp. This result can be attributed to the formation of hydrogen bonds with polyphenols and the presence of sugars in the chemical structure of these compounds, which improves their solubility [25]. The EtOH and WE extract also showed low levels of anthocyanins. Precipitation and formation of opaque sediments during the addition of the solvents were noted, an action that can be attributed to an irreversible reaction with the chemical structure of the anthocyanins. It is known that anthocyanins are dependent on several factors that can react reversibly or irreversibly such as pH, concentration, type of solvent, high temperature, and pigment structure, among others, and although some studies point out that the concentration of ethanol has a positive effect on the extraction efficiency in these pigments, it was observed that alcoholic extracts with a concentration of 50–70% had a significant drop in extraction [6], and in fact, the same profile was verified in this study.

Extracts from samples CA2 and CA9 obtained the best TMA contents using alternative solvents. The hydroethanolic extract sample WM, as a conventional solvent, presented around 40.61% and 46.26% more in relation to these samples. This sample also showed the highest total flavonoid content along with sample W75 in relation to the other samples, while the samples CA2 and CA9 exhibited the lowest flavonoid contents. This behavior can be explained due to the polarity of the molecules involved in the extraction, and also the sensitivity to pH changes. The phytochemical compounds present in the fruits can vary in different degrees of polarization, influencing the extraction efficiency [26]; the TMA content in the CA2 and CA9 samples may have been favored by the polarity of the carboxyl groups. However, in an acidic medium, the polarity of the carboxyl group decreases and tends to protonate, forming the carboxylic acid in its neutral form, which is less polar. Methanol is a solvent that has high polarity, with the ability to stabilize ions through the transfer of a proton, thus establishing a hydrogen bond [27], and in combination with water, which is an extremely polar solvent, may have influenced in obtaining higher solubilization rates and consequently greater extraction capacities in the content of anthocyanins and flavonoids.

However, it is known that the susceptibility of anthocyanins to pH can make them more stable in acidic solutions than alkaline ones due to their amphoteric character, because, at pHs established in the range of 1 to 2, the flavylium cation predominates favoring the extraction [23], and consequently, presents in the form of oxonium salts, with an intensely reddish color of anthocyanins. It is observed that with increasing acidity, the anthocyanin content declines, and similarly, the flavonoid content for samples CA2 and CA9 was the lowest. Possibly, the acidity caused by the acidified solvent triggered the destruction of cell membranes, dissolving their contents in the extracts [28,29]. In general, both concentrations of 2.4% and 9.6% of citric acid presented the best result for the extraction of TMAs compared to the alternative solvents studied.

Regarding the antioxidant action of extractor solvents, the DPPH scavenging activity of jambolan samples showed that the alternative solvents W25 and W75 had the best antioxidant capacity, being superior to conventional solvents EtOH, WM, and WE, yet differing significantly from each other (*p* < 0.05). There was a moderate positive correlation (R = 0.6735) with the TPCs, demonstrating that with the increase in the content of these compounds there is an increase in antioxidant activity. On the other hand, the alternative solvents from the W25, W50, and W75 samples showed the lowest antioxidant activity in the ABTS assay, while the conventional solvent WM showed the highest activity and with a strong negative correlation (R = −0.7738) with the content of phenolic compounds. Generally, the antioxidant action measured by the DPPH and ABTS assays is attributed to polyphenols with a high degree of hydroxylation, such as flavonoids and anthocyanins, which, by donating hydrogen atoms, can stabilize the free radicals formed [30]; however, as the levels of these TPCs increase, the antioxidant activity decreases, indicating that the activity of these extracts may be related to the content of TMAs or other unidentified compounds, such as alkaloids, carotenoids, and others.

The FRAP assay is performed to evaluate the total antioxidant power of experimental extracts, where it estimates the electron donating capacity of any compound based on the reduction of ferric ion (Fe^3+^, such as tripyridyl triazine ferric: Fe^3+^-TPTZ) to ferrous ion (Fe^2+^, as ferrous tripyridyl triazine: Fe^2+^-TPTZ) [31]. The results of the FRAP assay demonstrated that the WM conventional solvent sample exhibited the highest reducing power, with the highest TFs content, while the acidified samples CA2 and CA9 showed the weakest reducing power, and also the lowest TFs contents, confirming that there was a strong positive correlation (R = 0.8353) with the total flavonoids content; that is, as the contents of these compounds increased, their antioxidant capacity increased. Some studies indicate that anthocyanins also have free radical scavenging properties, high nitric oxide scavenging power, ferric and hydroxyl reducing power, as well as superoxide radical scavenging activities [32]; however, in this work, there was no correlation between total anthocyanins and neither with TPCs in relation to the FRAP assay.

The concentrated bioactive compounds of jambolan extracts are shown in Figure 1. There was an increase in all concentrated compounds in relation to jambolan extracts. The highest levels of TPCs and TFs were identified in the W25 concentrated sample with an increment of 361.39% and 3759.37%, respectively, in relation to the W25 extract sample and in the WM concentrated sample with an increase of 631.21% and 3032.14%, respectively, in relation to the WM extract sample.

The highest TMAs contents were identified in the W25 concentrated sample with an increment of 1228.25% in relation to the W25 extract sample and in the WE concentrated sample with an increment of 2431.18% in relation to the WE extract sample.

Unlike the extracts, in this concentration step, TMAs levels were more evident in the W25 concentrated sample and lower in the W50 and W75 concentrated samples compared to their extracts. Possibly, the subsequent exposure to drying (70 °C and 20 min) of the samples may have impaired the decomposition of biologically active substances, which is especially visible in the case of anthocyanins. Kowalska et al. [6] visualized in their studies that small variations in the concentration of anthocyanins obtained in the extraction systems used (for water and water–glycerol mixtures) between the temperature of 50 °C and 80 °C could be related to the progressive degradation of the isolated anthocyanins. Likewise, the increase in temperature may have favored the reaction speed and transfer rates of phenolic and flavonoid compounds, since they easily oxidize when exposed to sunlight and oxygen, generating radicals that can react with other radicals to form dimers.

With the concentration of the extracts, the concentrated samples also had an increase in their antioxidant capacity (Figure 2) when compared with the tests of jambolan extracts. The W25 concentrated samples showed the highest antioxidant activity in the DPPH and FRAP assays, while the concentrated WM and EtOH concentrated samples showed the highest antioxidant activity by the ABTS method. On the other hand, concentrated samples CA2 and CA9 showed the lowest antioxidant activity in the three assays performed. There was a strong positive correlation between total phenolic compounds, total flavonoids, and antioxidant capacity. Regarding TPCs, the ABTS test showed a strong and positive correlation, with a Pearson’s coefficient of R = 0.9455; for the FRAP assay, the coefficient was R = 0.8751, (*p* < 0.05); regarding TFs, there was a moderate positive correlation with the DPPH assay, R = 0.6087, and strong correlations for the ABTS assay, R = 0.7580 and FRAP, R = 0.9881, (*p* < 0.05), confirming that antioxidant activity is related with the presence of these compounds in samples of jambolan concentrates.

Validating an appropriate method to determine the antioxidant capacity and bioactive compounds depends on the interactions between the extraction solvents and the compositional matrix of the evaluated fruit. This demonstrates that the choice of the most suitable solvent for the extraction of bioactive compounds from jambolan depends on the interactive chemical components, such as functional group, pigment chain length, in addition to nutritional composition [33].

In summary, the DPPH method was better evidenced in the aqueous extract and concentrate at W25 and W25, the ABTS method showed greater activity in the alcoholic, hydromethanolic, and hydroalcoholic extracts and concentrates, while the FRAP method showed greater activity for the extracts EtOH, WM, WE, and concentrates W25 and WM. For the extraction of bioactive compounds, the use of water at 75 °C as an alternative solvent obtained the highest content of total phenolic compounds. Despite not reaching the same total flavonoid content as the jambolan pulp, the WM and W75 extracts had the highest contents compared to the other samples and did not differ from each other (*p* < 0.05), demonstrating that water at 75 °C can be used with solvent. For the extraction of TMAs, the use of acidified water 2.4% and 9.6% demonstrated more efficiency as an alternative solvent, as well as W25 extract proved to be more efficient than the EtOH and WE extracts, indicating that it can be a more efficient, ecological substitute, and low-cost compared to traditional solvents.

### 2.3. Anthocyanins by UPLC-MS/MS

Anthocyanins in jambolan extracts and concentrate samples were also evaluated using UPLC-MS/MS (Figure 3). The identified compounds included delphinidin 3,5-diglucoside (*m*/*z* 627.0), cyanidin 3,5-diglucoside (*m*/*z* 611.0), delphinidin 3-glucoside (*m*/*z* 465.0), petunidin 3,5-diglucoside (*m*/*z* 461.0), cyanidin 3-glucoside (*m*/*z* 449.0), peonidin 3,5-diglucoside (*m*/*z* 625.0), malvidin 3,5-diglucoside (*m*/*z* 655.0), petunidin 3-glucoside (*m*/*z* 479.0), and malvidin 3-glucoside (*m*/*z* 493.0).

Anthocyanins are important phenolic compounds found in the jambolan fruit, which typically has a colorless pulp. The detection of anthocyanins in this part of the fruit may be due to the migration from the peel and/or colored seed, most likely during the preparation of the sample or as a result of excessive ripening [34].

In general, the highest molar proportions were for petunidin 3,5-diglucoside, not detected in the concentrated sample CA9. Malvidin 3,5-diglucoside, delphinidin 3,5-diglucoside, and cyanidin 3,5-diglucoside can be highlighted when compared to the other samples. Koop et al. [35] also presented high molar concentrations of delphinidin 3,5-diglucoside, petunidin 3,5-diglucoside, and malvidin 3,5-diglucoside in the jambolan extract.

Samples with water as W25 and W50 showed good scores, which may be due to the following: the use of water as a solvent, since anthocyanins are water-soluble pigments [36]; and the low extraction temperature, as anthocyanins are unstable compounds, which can be preserved at mild/low temperatures [37]. The results also strongly support the notion that the structure of the glycosylated anthocyanin affects its stability: the 3-glycoside anthocyanins were more unstable than their respective 3,5-diglucoside derivatives, especially cyanidin 3-glucoside, i.e., B-ring di-substituted non-methoxylated anthocyanins [38].

The WM extract presented the second-highest molar proportions of the compounds petunidin 3,5-diglucoside, malvidin 3,5-diglucoside, delphinidin 3,5-diglucoside, and cyanidin 3,5-diglucoside. It also presented the highest molar concentrations of petunidin 3-glucoside, delphinidin 3-glucoside, malvidin 3-glucoside, and cyanidin 3-glucoside. These compounds were identified and described previously for this fruit [34]. The increase in monoglycoside compounds can be explained by the lysis of the cell membrane caused by the extraction with methanol and water, resulting in simultaneous dissolution and stabilization of the pigmented groups of anthocyanin [39].

According to the heatmap, some anthocyanins were not identified in the samples WE (extract), CA9 (extract and concentrated), and W50 (extract). The anthocyanins delphinidin 3-glucoside, peonidin 3,5-diglucoside, and malvidin 3-glucoside were not identified in the WE extract. In the CA9 extract, cyanidin 3,5-diglucoside, delphinidin 3-glucoside, and cyanidin 3-glucoside were not identified. In the CA9 concentrated sample, delphinidin 3-glucoside and malvidin 3-glucoside were not found. Finally, in the W50 extract, cyanidin 3-glucoside, peonidin 3,5-diglucoside, and petunidin 3-glucoside were not detected. The difference in the profile of anthocyanins is related to the capacity of the specific solvent to absorb and extract anthocyanins, as well as the solvent:sample ratio and the extraction temperature [40].

These current results confirmed that the jambolan fruit is an excellent source of bioactive anthocyanins and strongly suggest that its undervalued application in food should be reconsidered [34].

## 3. Materials and Methods

### 3.1. Chemicals and Reagents

The solutions were prepared using analytical reagents and Milli-Q water for the required analyses. Reagents for antioxidants (Trolox, ABTS, TPTZ, DPPH, Folin–Ciocalteu reagent), were obtained from Sigma-Aldrich Chemical Co. (St. Louis, MO, USA). All reagents/solvents were of analytical grade in accordance with the requirement.

### 3.2. Source of Fruits and Sample Preparation

Jambolan (*Syzygium cumini* (L.) Skeels) in natura was purchased at Santa Mônica farm in Rolândia, Southern Region of Brazil (23°18′38″ S; 51°22′10″ W). The jambolan fruits were harvested from the January 2022 harvest, in a single batch of five kilos. After the acquisition, the fruits were selected at the stage of complete physiological maturation (ripe and purple), classified, washed with drinking water, sanitized with sodium hypochlorite (150 mg L^−1^) by immersion for 15 min, rinsed again, and dried on absorbent paper. The seeds were manually removed and the skin and pulp were homogenized in a blender (Oster, 750W, Balneário Piçarras, Brazil), without adding water. Jambolan pulp was characterized in the relation of chemical composition and physical–chemical parameters. Subsequently, it was vacuum-filtered (Prismatec, model 131, 2VC, Itu, Brazil) and the supernatant was characterized for the content of bioactive compounds and antioxidant capacity. Then, eight samples were separated and identified for subsequent solvent extraction steps.

### 3.3. Extraction and Concentration

For sustainable extraction, different conditions and solvents at different concentrations were considered (Table 3). Conventional solvents used were ethanol as well as binary mixtures of ethanol:water and methanol:water. Alternative solvents were water at different temperatures (25 °C, 50 °C, and 75 °C) and citric acid at different concentrations (2.4% and 9.6%). The extraction was performed with 10 g of jambolan pulp, with a sample-to-solvent ratio of 1:3 (*w*/*v*) under constant stirring for 30 min at 150 rpm in a mechanical stirrer (Fisatom, Model 713, São Paulo, Brazil).

The solvents used and their parameters (temperature and concentration) were based on previous studies. Water baths were used to maintain the temperature to extract the samples with water at 25 °C, 50 °C, and 75 °C and the samples with conventional solvents at 25 °C. After the period, the extracts were filtered under vacuum conditions (Prismatec, model 131, 2VC, Itu, Brazil). The concentration of the newly prepared extracts was carried out with 20 mL of each extract, filtered through 3 × 2” sieves (Abronzinox, ABNT 200/TYLER, São Paulo, Brazil) with an opening of 74 µm and concentrated in a rotary evaporator (Buchi, RE 120, New Castle, DE, USA) until dry for 25 min at 70 °C. After concentration, the samples were resuspended with 3 mL of distilled water, identified, and frozen at −18 °C for subsequent analysis.

### 3.4. Jambolan Pulp Characterization

Centesimal composition: The analyses of moisture, ash, and protein content were in agreement with IAL [41]. Lipid determination was performed by cold solvent extraction [42] and quantification was performed by difference: % carbohydrate = (100 − protein + lipid + moisture + ash).

Physical–chemical analysis: Jambolan pulp was still analyzed for total titratable acidity, soluble solid content (digital refractometer—HI 96801 Refractometer), and pH [41]. The activity of water (aw) was carried through in a Aqualab^®^ equipment (Braseq^®^, Aqualab, Jarinu, Brazil) 25 °C.

### 3.5. Analysis of Bioactive Compounds

#### 3.5.1. Total Phenolic Compounds (TPCs)

The samples were determined by Folin–Ciocalteu [43]. Absorbance was verified in a spectrophotometer (Bel UV-Vis, model Uv-m51, Piracicaba, Brazil) at 725 nm. Gallic acid was used as a standard for the calibration curve. The results were expressed in mg of gallic acid equivalent (GAE)/100 g) of the product and the calibration curve was as follows: y = 0.0055x + 0.06; R^2^ = 0.9957.

#### 3.5.2. Total Flavonoids (TFs)

The samples were analyzed in a colorimetric assay using aluminum chloride (AlCl_3_), sodium nitrite (NaNO_2_), and sodium hydroxide (NaOH) [44]. Absorbance was immediately verified in a spectrophotometer (Bel UV-Vis, model Uv-m51, Piracicaba, Brazil) at 510 nm. The calibration curve was prepared using a standard quercetin solution and the results were expressed in mg of quercetin equivalent (QE)/100 g of product and the calibration curve was as follows: y = 0.0014x + 0.0697; R^2^ = 0.9904.

#### 3.5.3. Total Monomeric Anthocyanins (TMAs)

The differential pH method [45] will be used to determine total monomeric anthocyanins, with potassium chloride (KCl) and sodium acetate (C_2_H_3_NaO_2_) reagents. The absorbance will be verified in a spectrophotometer (Bel UV-Vis, model Uv-m51, Piracicaba, Brazil) at 520 and 700 nm after 20 min of incubation at 25 °C. The results will be expressed in mg cyanidin-3-glucoside (c-3-g)/100 g of the product, according to Equation (1):Anthocyanin pigment = (A × MW × Df × 10^3^)/ε × λ(1)
where: A is (ABS 520 nm–ABS 700 nm) pH 1.0–(ABS 520 nm–ABS 700 nm) pH 4.5; MW is 449.2 g mol^−1^ (molar mass of c-3-g); Df is the dilution factor; 10^3^ is the factor for conversion from g to mg; *ε* is 26,900 L mol^−1^ cm^−1^ (molar extinction coefficient, for c-3-g); and *λ* is 1 cm (cuvette optical pathlength).

#### 3.5.4. Antioxidant Activity

##### Ferric Reducing Antioxidant Power (FRAP)

It was evaluated using TPTZ reagents (2,4,6-tris(2-pyridyl)-s-triazine), a 0.3 M acetate buffer, and 20 mM ferric chloride [46]. Absorbance was verified at 595 nm and the readings were performed in a spectrophotometer (Bel UV-Vis, model Uv-m51, Piracicaba, Brazil). The results were expressed in µM eq Fe_2_SO_4_/g product and the calibration curve was as follows: y = 0.0016x – 0.0368; R^2^ = 0.9901.

##### DPPH Assay

Antioxidant activity (DPPH—2,2-diphenyl-1-picrylhydrazyl) was determined by the colorimetric method at 515 nm [46] and the readings performed in a spectrophotometer (Bel UV-Vis, model Uv-m51, Piracicaba, Brazil). Trolox was used as a standard for the calibration curve, the results were expressed in µM Trolox Equivalent (TE)/g product, and the calibration curve was as follows: y = 0.1101x + 3.8612; R^2^ = 0.9979.

##### ABTS Assay

The method was evaluated using a colorimetric assay with the ABTS reagents [2,2-azinobis (3-ethylbenzothiazoline-6-sulfonic acid)]) and potassium persulfate (K_2_S_2_O_8_) [11]. Absorbance was verified at 734 nm after 6 min of incubation at 25 °C and the readings performed in a spectrophotometer (Bel UV-Vis, model Uv-m51, Piracicaba, Brazil). A calibration curve was prepared using a standard Trolox solution, the results were expressed in µM Trolox Equivalent (TE)/g product, and the calibration curve was as follows: y = −0.0003x + 0.6889; R^2^ = 0.9925.

#### 3.5.5. Identification of Anthocyanins by UPLC-MS

A UPLC Acquity H-CLASS coupled with a Xevo TQD triple quadrupole mass spectrometer equipped with a Z spray ™ ESI interface operating in positive and negative mode was used to perform the chromatographic analysis. Chromatographic separation was carried out using a Waters Acquity UPLC ^®^ BEH C18 (Waters Corp., Milford, CT, USA) column with 1.7 µm particles (50 × 2.1 mm id) at a flow rate of 0.150 mL min^−1^. The column was maintained at 30 ± 1 °C, and the sample injection volume was 1.5 µL. The mobile phase consisted of (A) water (acidified with 0.1% formic acid) and (B) methanol (MeOH). A gradient elution was used, and the percentage of organic solvent (MeOH) was changed linearly as follows: 0–0.01 min (10% B), 1 min (30% B), 1.5 min (40% B), 2 min (50% B), 4–7 min (60% B), 7.5 min (50% B), 8 min (30% B), and 8.5–13 min (10% B). The mass spectrometer was operated using an electrospray ionization (ESI) source in negative mode. The ESI conditions were as follows: capillary voltage, 3.0 kV; extractor voltage, 3.0 V; source temperature, 130 °C; desolvation gas temperature, 550 °C; cone gas flow rate (nitrogen) of 50 L h^−1^; and desolvation gas flow rate (also nitrogen) of 700 L h^−1^. Argon (99.9%) from White Martins (Rio de Janeiro, Brazil) was used as the collision gas at a constant pressure of 3.00 × 10^−3^ mbar. The mass spectrometer was operated in MS/MS mode using selected reaction monitoring (SRM). The most intense ion transition was selected for quantification, and the second-most for qualification. Specific MS/MS parameters were set for each anthocyanin. MassLynx and QuanLynx software version 4.1 (Waters) were used for instrument control and data processing [47].

### 3.6. Statistical Analysis

All analyses were performed in triplicate and submitted to statistical analysis of variance and Tukey’s test for the least significant difference (*p* < 0.05) using the Sisvar 5.6 program. The Pearson correlation was also calculated in some analyses, for comparison. Calibration curves for the antioxidant analyses were performed using the GraphPrism 5 program.

## 4. Conclusions

Jambolan fruit is a rich source of phytochemical compounds, with anthocyanins, the bioactive compound with recognized antioxidant action, soluble in aqueous mixtures, and which are responsible for its highly attractive color being feasible in uses in numerous food formulations. The use of a simple process with alternative solvents, which are cheap, non-toxic, accessible, and safe, demonstrated high efficiency in the extraction of bioactive compounds with antioxidant action from this fruit.

Water, with its stronger polarity compared to conventional solvents such as methanol and ethanol, can serve as an alternative extraction solvent for phenolic compounds, total flavonoids, and total anthocyanins. For water, the optimal temperature for extraction of phenolic compounds and TFs is 75 °C, while for anthocyanins, it is 25 °C. Citric acid 2.4% has proven to be the most efficient alternative solvent when compared to ethanol. A systematic study for the application and stability of bioactive compounds from jambolan in food or pharmaceutical products is recommended, since this fruit has high amounts of phytochemical compounds that may suffer interference from external variables, affecting the characteristics.

## Figures and Tables

**Figure 1 plants-13-02065-f001:**
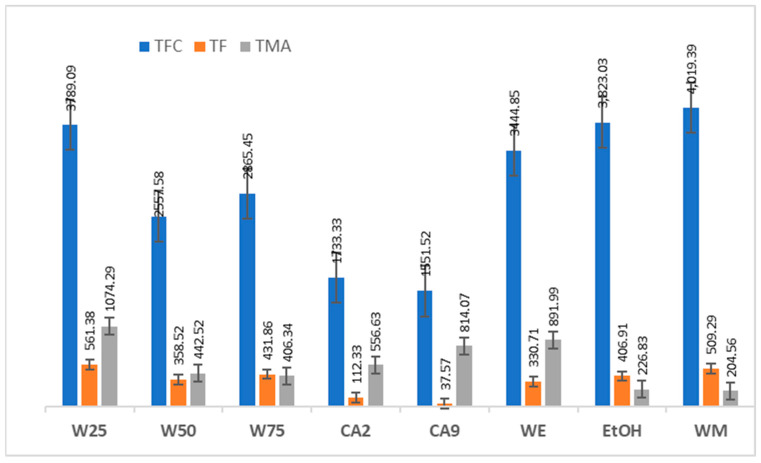
Total phenolic compounds (TPCs), total flavonoids (TFs), and total monomeric anthocyanins (TMAs) of jambolan concentrates extracts using different solvents.

**Figure 2 plants-13-02065-f002:**
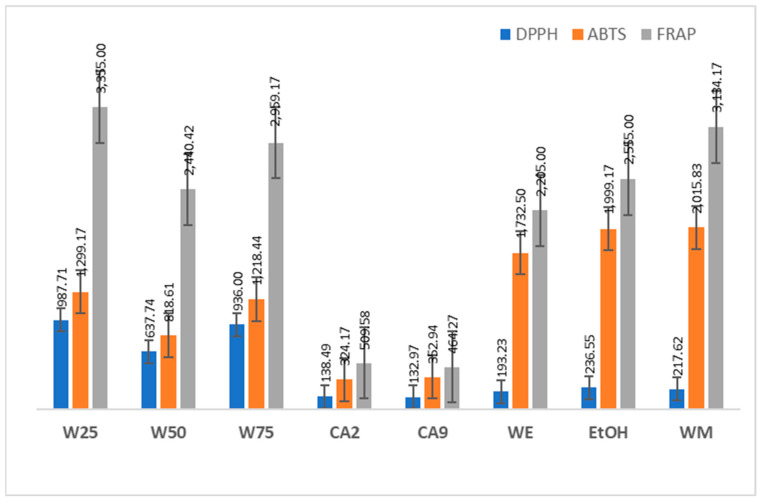
Antioxidant activity of jambolan concentrates extracts using different solvents.

**Figure 3 plants-13-02065-f003:**
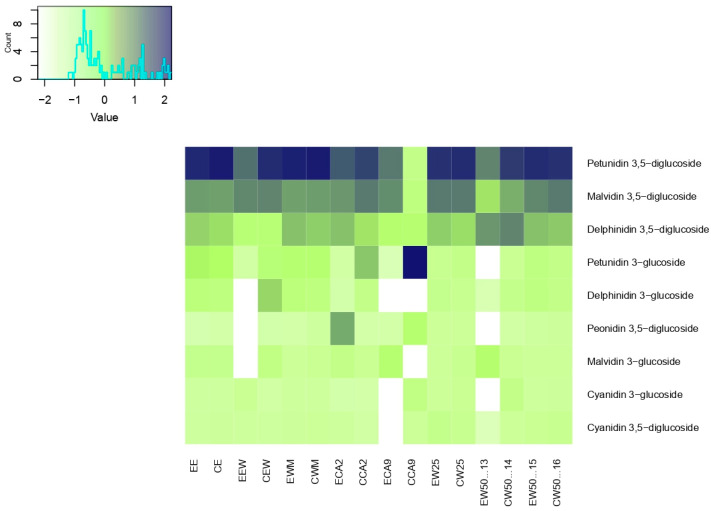
Anthocyanins in jambolan extracts and concentrate samples evaluated by heatmap, where the initial letter E corresponds to extract and C to concentrate. The other abbreviations correspond to the solvent used and the variables of temperature and concentration, as seen in Table 3.

**Table 1 plants-13-02065-t001:** Means values for analysis of the composition of jambolan pulp.

Proximate Composition and Physicochemical Parameters		Bioactive Compounds and Antioxidant Activity	
Parameter	Mean (Wet Basis)	Parameter	Mean (Wet Basis)
Moisture (g/100 g)	83.76 ± 0.45	TPCs	711.51 ± 7.57
Carbohydrates (g/100 g)	14.68 ± 0.42	TFs	294.97 ± 4.59
Proteins (g/100 g)	0.65 ± 0.01	TMAs	35.57 ± 6.47
Lipids (g/100 g)	0.45 ± 0.01	FRAP	683.02 ± 9.83
Ashes (g/100 g)	0.46 ± 0.01	DPPH	158.69 ± 3.20
Soluble solids (°Brix)	13.30 ± 0.02	ABTS	98.97 ± 4.16
pH	3.76 ± 0.01		
Aw	0.98 ± 0.01		
Acidity (g of citric acid /100 g)	0.88 ± 0.01		

Data are expressed as means ± standard error (*p* ≤ 0.05; Tukey’s test). TPCs (total phenolics compounds expressed in mg GAE/100 g); TFf (total flavonoids expressed in mg QE/100 g); TMAs (total monomeric anthocyanins expressed in mg c-3-g/100 g); DPPH and ABTS expressed in μM TE/g and FRAP expressed in μM eq Fe_2_SO_4_/g.

**Table 2 plants-13-02065-t002:** Total phenolic compounds (TPCs), total flavonoids (TFs), total monomeric anthocyanins (TMAs), and antioxidant activity of jambolan extracts using different solvents.

Samples	TPCs (mg GAE/100 g)	TFs (mg QE/100 g)	TMAs (mg c-3-g/100 g)	DPPH (μM TE/g)	ABTS (μM TE/g)	FRAP (μM eq Fe_2_SO_4_/g)
W25	821.21 ^c^ ± 11.69	14.52 ^b,c^ ± 0.86	80.88 ^c^ ± 7.55	205.91 ^a^ ± 7.47	44.52 ^e^ ± 5.87	113.53 ^c^ ± 1.18
W50	1262.42 ^b^ ± 21.27	12.28 ^c^ ± 1.71	21.65 ^d^ ± 5.12	152.12 ^b^ ± 4.52	25.41 ^g^ ±1.07	80.53 ^e^ ± 3.63
W75	1347.27 ^a^ ± 20.65	19.59 ^a^ ± 1.05	24,32 ^d^ ± 7.73	207.06 ^a^ ± 0.92	34.86 ^f^ ± 2.14	92.37 ^d^ ± 1.61
CA2	528.48 ^f^ ± 27.52	7.78 ^d^ ± 0.57	110.04 ^b^ ± 8.26	106.95 ^e^ ± 4.35	64.63 ^c^ ± 2.91	73.72 ^e^ ± 2.97
CA9	655.15 ^e^ ± 2.80	1.57 ^e^ ± 0.14	99.58 ^b^ ± 4.88	115.08 ^d,e^ ± 1.98	54.85 ^d^ ± 1.35	30.36 ^f^ ± 0.45
WE	738.78 ^d^ ± 16.89	13.95 ^b,c^ ± 0.28	35.23 ^d^ ± 1.86	122.97 ^c,d^ ± 1.99	86.86 ^a,b^ ± 2,71	137.70 ^b^ ± 4.63
EtOH	744.24 ^d^ ± 24.20	16.26 ^b^ ± 0.72	25.38 ^d^ ± 6.95	159.20 ^b^ ± 2.21	81.19 ^b^ ± 2,69	133.53 ^b^ ± 3.59
WM	549.69 ^f^ ± 3.75	21.78 ^a^ ± 0.86	185.30 ^a^ ± 8.86	126.01 ^c^ ± 0.54	90.75 ^a^ ± 1,07	157.78 ^a^ ± 1.42
Mean	830.91 ± 10.46	13.47 ± 0.52	72.80 ± 2.95	149.41 ± 2.02	60.38 ± 1.73	102.44 ± 1.69
CV (%)	2.18	6.77	7.01	2.32	4.98	2.86

Data are expressed as means ± standard deviation (*n* = 3). Means with different letters in the same column are significantly different for *p* ≤ 0.05; Tukey’s test. W25 (water at 25 °C); W50 (water at 50 °C); W75 (water at 75 °C); CA2 (water with citric acid at 2.4%); CA9 (water with citric acid at 9.6%); WE (water with ethanol at 50%); EtOH (ethanol); WM (water with methanol at 50%).

**Table 3 plants-13-02065-t003:** List of samples obtained according to the solvent used and the variables of temperature and concentration.

Samples	Solvent	Temperature	Concentration
EtOH	Ethanol	25 °C	99.99% *v*/*v*
EW	Ethanol + water	25 °C	50:50% *v*/*v*
WM	Methanol + water	25 °C	50:50% *v*/*v*
CA2	Citric acid + water	25 °C	2.4:97.6% *m*/*v*
CA9	Citric acid + water	25 °C	9.6:90.4% *m*/*v*
W25	Water	25 °C	100% *v*/*v*
W50	Water	50 °C	100% *v*/*v*
W75	Water	75 °C	100% *v*/*v*

## Data Availability

The datasets supporting the conclusions of this article are included within the manuscript.

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
