# Peer review of "Extraction of Bioactive Compounds from the Fruits of Jambolan (Syzygium cumini (L.)) Using Alternative Solvents"

_plants, 2024, doi:10.3390/plants13152065_

Round 1

Reviewer 1 Report

Comments and Suggestions for Authors

An important task of green chemistry is to transform extraction methods in an environmentally friendly way by using solvents and solvent mixtures that are effective and do not pollute the environment. From this point of view, the choice of the topic of the manuscript is very important.

Some observations:

In many cases, the measured values ​​can be found in several places in the article, table, text, or figure. This is unnecessary, it is enough to enter the value in one place and refer to it later.

Figure 3 contains too much data, the colors are also very similar, it is difficult to see through, I recommend changing this.

When preparing the concentrates, was the water content of the final concentrates the same? Only the time and method (25 minutes at 70ºC.) of evaporation of the concentrates were given, which results in a different final water content product for different solvents.

Author Response

The information is in the attached file.

Reviewer 2 Report

Comments and Suggestions for Authors

The propose study deals with the development of extracting procedure aiming at substituting organic solvents.

This study is however poorly written with too many abbreviations and numerous too long sentences. A general re-written version must be proposed.

Author Response

(The authors gave the same response as above.)

Reviewer 3 Report

Comments and Suggestions for Authors

Dear Authors,

This work “Jambolan (Syzygium cumini (L.) Skeels): extracts obtained through sustainable extraction

The article is written quite well. Despite the small number of methods used, the work is of real practical interest. Nowadays, there is a lot of information on the functional properties of compounds of plant origin. Therefore, eco-friendly extraction methods for optimal use of plants in the cosmetic, pharmaceutical, or food industries are of paramount importance.

I have a few comments regarding the content:

1.      The title looks like the title of a review article. In addition, it would be good to clarify in the title which group of compounds was studied (phenolic compounds).

2.      Key wards are confused. Add Syzygium cumini (L.) Skeels (and in abstract); jambolan; anthocyanins.

3.      The system of abbreviations used makes it very difficult to understand. Especially in the abstract. To understand the essence of the results, you need to scroll through and delve into the materials and methods. It would be nice to include a table with the meanings of abbreviations in the results (2.2.). The letter J in the abbreviations is clearly superfluous, since no other plants were used.

4.      Chapter 2.1 is written as if the authors did not conduct their own analysis but misrepresented their review of the available data. Rewrite the chapter so that it is clear what the authors have done and what is being discussed based on already-published data.

5.      Table 2. CV% and mean are unclear. What is it?

6.      The result of concentration remains unclear. It was 30 ml; how many ml came out after concentration? Figure 1 is also incomprehensible: how did you carry out statistical analysis without mean and standard error? Was there only one repetition? Why didn’t the authors compare the content of active components in the original and concentrated samples?

7.      Figure 2 is also incomprehensible: how did you carry out statistical analysis without mean and standard error?

8.      Figure 3 is completely unreadable. Better make a heatmap.

Best regards

Author Response

(The authors gave the same response as above.)

Round 2

Reviewer 2 Report

Comments and Suggestions for Authors

Thanks to the authors to have considered improvement suggests.

Still minor corrections:

- L82: Table 1 describes

L202: "... carboxyl groups. However, ..."

- L419: AlCl3

Author Response

Dear Editor, please find enclosed the revised manuscript entitled “Extraction of bioactive compounds from the fruits of Jambolan (Syzygium cumini (L.)) using alternative solvents”, which we intend to publish in the journal Plants. The manuscript was revised by taking the referee’s comments into account and changes are highlighted in red. We would like to acknowledge the careful evaluation of the manuscript and the suggestions for improving our work.

We look forward to seeing our manuscript published in Plants.

Yours sincerely,

Grasiele Scaramal Madrona.

Reviewer(s)' Comments to Author:

#Reviewer: 1

No comment.

#Reviewer: 2

Thanks to the authors to have considered improvement suggests.

Still minor corrections:

- L82: Table 1 describes

The change was performed in the manuscript.

L202: "... carboxyl groups. However, ..."

The change was performed in the manuscript.

- L419: AlCl3

The change was performed in the manuscript.

#Reviewer: 3

No comment.

We are thankful for the considerations, and please, do not hesitate to contact us if you have any query.

Sincerely Yours,

                                                        Grasiele Scaramal Madrona.

                                               State University of Maringá - BRAZIL
